# Carnosine to Combat Novel Coronavirus (nCoV): Molecular Docking and Modeling to Cocrystallized Host Angiotensin-Converting Enzyme 2 (ACE2) and Viral Spike Protein

**DOI:** 10.3390/molecules25235605

**Published:** 2020-11-28

**Authors:** Loai M. Saadah, Ghina’a I. Abu Deiab, Qosay Al-Balas, Iman A. Basheti

**Affiliations:** 1Faculty of Pharmacy, Applied Science Private University, 11931 Amman, Jordan; dr_iman@asu.edu.jo; 2School of Pharmaceutical Sciences, Universiti Sains Malaysia, Penang 11800, Malaysia; 3Faculty of Pharmacy, Yarmouk University, 21163 Irbid, Jordan; ghinaadeiab@gmail.com; 4Faculty of Pharmacy, Jordan University for Science & Technology, 22110 Irbid, Jordan; qabalas@just.edu.jo; 5Faculty of Pharmacy, The University of Sydney, Sydney 2006, Australia

**Keywords:** COVID-19, carnosine, angiotensin-converting enzyme 2 (ACE2), practitioner, molecular docking, modeling

## Abstract

Aims: Angiotensin-converting enzyme 2 (ACE2) plays an important role in the entry of coronaviruses into host cells. The current paper described how carnosine, a naturally occurring supplement, can be an effective drug candidate for coronavirus disease (COVID-19) on the basis of molecular docking and modeling to host ACE2 cocrystallized with nCoV spike protein. Methods: First, the starting point was ACE2 inhibitors and their structure–activity relationship (SAR). Next, chemical similarity (or diversity) and PubMed searches made it possible to repurpose and assess approved or experimental drugs for COVID-19. Parallel, at all stages, the authors performed bioactivity scoring to assess potential repurposed inhibitors at ACE2. Finally, investigators performed molecular docking and modeling of the identified drug candidate to host ACE2 with nCoV spike protein. Results: Carnosine emerged as the best-known drug candidate to match ACE2 inhibitor structure. Preliminary docking was more optimal to ACE2 than the known typical angiotensin-converting enzyme 1 (ACE1) inhibitor (enalapril) and quite comparable to known or presumed ACE2 inhibitors. Viral spike protein elements binding to ACE2 were retained in the best carnosine pose in SwissDock at 1.75 Angstroms. Out of the three main areas of attachment expected to the protein–protein structure, carnosine bound with higher affinity to two compared to the known ACE2 active site. LibDock score was 92.40 for site 3, 90.88 for site 1, and inside the active site 85.49. Conclusion: Carnosine has promising inhibitory interactions with host ACE2 and nCoV spike protein and hence could offer a potential mitigating effect against the current COVID-19 pandemic.

## 1. Introduction

Confirmed novel coronavirus disease (COVID-19) was first reported to the World Health Organization (WHO) in December 2019 and is probably the most challenging out of the previous coronavirus infections [1,2,3]. COVID-19 has evolved since 30 January 2020, into an unprecedented worldwide pandemic [4]. By the time of writing this paper (17 October 2020), confirmed cases have reached close to 40 million people with more than 1.1 million deaths globally [5]. Therefore, researchers and practitioners are under pressure to repurpose, identify, and develop new drugs for this insurgent healthcare emergency [6].

In COVID-19, coronavirus (known as SARS-CoV-2) uses ACE2 for the entry into host epithelial and lung cells. This is a similar theme to the older systemic respiratory distress syndrome coronavirus (SARS-CoV) [7,8]. However, SARS-CoV-2 seems to fit ACE2 better than SARS-CoV [9]. Obviously, nCoV spike protein is responsible for this improved fitness [10,11]. Additionally, most patients have cardiovascular disease, hypertension, and diabetes and the ACE2 is upregulated in these patients [12]. Therefore, short term inhibition of ACE2 interaction with the viral spike protein would actually benefit COVID-19 patients. It will also prevent the lethal virus from utilizing this upregulated host enzyme to gain entry and replicate into human cells. Type 2 transmembrane serine protease (TMPRSS2) is another, more extensively studied, important host target that facilitates coronavirus to enter into human lung cells [13]. As a result, a drug that can interfere with ACE2 and/or TMPRSS2 interaction with the virus should, in fact, help in combating COVID-19. Due to a more extensive evaluation of TMPRSS2 and SARS-CoV-2 proteases, ACE2 seems to be a relatively good, less evaluated, starting point for potential COVID-19 therapies [14].

Now, to begin the search for an ACE2-mediated solution to the current pandemic, one should initially look at drugs with structures that can inhibit this enzyme. It is now possible to find a general structural scaffold for ACE2 inhibitors [15]. On the other hand, the concept of ACE2 inhibitors is still under development, and therefore, a simple similarity or diversity search on approved or investigational drugs matching the general ACE2 scaffold would most likely yield few, if any, candidates to test. Similarity or diversity search is a relatively new concept in chemo-informatics and there are numerous methods or equations to study it [16]. These include Euclidean, Manhattan, and Mahalanobis metrics [17]. However, Tanimoto index is the simplest and most direct such distance measure which calculate the fraction shared bits between chemical fingerprints (also known as pharmacophores) [18]. Using chemical similarity search we can then most likely find few new drugs to repurpose to COVID-19. Finally, researchers can perform molecular docking and modeling on the identified molecules. In the docking and modeling, our team employed two strategies. First, compare interaction of the identified molecules docked to the host ACE2 cocrystallized with SARS-CoV-2 spike protein to known inhibitors of this target. Second, assess whether the identified molecule likes to interact more with the protein–protein structure rather than the ACE2 active site.

## 2. Aims

The aim of this study was to identify currently approved or known experimental compounds as probable ACE2 inhibitors. These were then evaluated with molecular docking and modeling to the ACE2 with the viral spike protein to help determine if they can potentially inhibit this protein–protein interaction.

## 3. Methods

Figure 1 shows an overview of the methods used in this study. As a starting point for new compounds with an inhibitory effect against ACE2, a simple Google search was performed to elucidate the SAR of ACE2 inhibitors.

This led us to the scaffold general structure of ACE2 inhibitors that has been reported by Dales and Torres (Figure 2) [15,19]. We considered the simplest chemical structure possible where R^1^ and R^3^ groups are replaced with hydrogen atoms, and R^2^ is dropped (Figure 2).

Next, as angiotensin-converting enzyme inhibitors (ACEI) work on the first known member in this enzyme family we randomly selected eight and compared to the general structure of ACE2 inhibitor. This step could enable us to distinguish if any ACEI would preferentially be more cross active to ACE2 (Figure 3). Evaluated ACEI included captopril, enalapril, ramipril, trandolapril, perindopril, benazepril, fosinopril, and temocapril. However, a full discussion of the relationship of ACE1 to ACE2 is beyond the scope of the current work.

Moreover, molinspiration bioactivity score calculator was utilized to predict the activity of the identified drug candidates at the six important ligands for ACE2; namely, 1: G protein coupled receptor (GPCR) ligand, 2: ion channel modulator, 3: kinase inhibitor, 4: nuclear receptor ligand, 5: protease inhibitor, and 6: enzyme inhibitor [20]. On the other hand, ChemMine similarity scoring was used to check how close the molecule to the general ACE2 inhibitor scaffold structure was [21]. Obviously, similarity metrics, atom pair (AP) Tanimoto method, maximum common substructure (MCS) Tanimoto method, MCS Min, and MCS Max, were compared for the various agents. The higher values the of these metrics, the closer the drug is to the general scaffold ACE2 inhibitor. ChemSpider and DrugBank were utilized to look for new compounds similar to the scaffold general structure of ACE2 inhibitors [22,23]. As we will see, carnosine emerged as the best-known ligand matching the general scaffold ACE2 inhibitor. Bioactivity scoring was made on every new potential or reference molecule identified. Absolute differences between the binding scores of each molecule and the general scaffold structure were calculated. The differences for each of the six ligands were, then, summed up into a final total number. Hence, the lower this absolute summed total the closer supposedly the identified drug is to the general scaffold structure.

Preliminary molecular docking was performed using SwissDock server as described by other research groups [24]. ACE2 code in SwissDock server is 6M0J. Note this is the cocrystallized structure of ACE2 with nCoV spike protein. Subsequently, reference compounds were used during docking to ACE2 including ACE1 inhibitor, enalapril, and melatonin (Figure 4). While enalapril is an ACEI, melatonin has shown good docking results at ACE2 (Figure 4) [25]. Binding modes of carnosine were visually presented with University of California San Francisco (UCSF) chimera version 1.14. To compare docking results of the various drugs both the lowest estimated Gibbs free delta G (ΔG) energy for cluster 0 first elements and the summary of all binding modes were considered.

Next, more detailed molecular docking and modeling was pursued. Crystal structure of ACE2 bound with nCoV spike protein was deposited in protein data bank (PDB) under the code (2AJF) [26]. Authors prepared 2AJF with the “prepare protein” protocol (Discovery studio 2020 from Biovia^®^, San Diego, CA, USA). Preparation parameters were set to default. This protocol will standardize atom names, insert missing atoms in residues, and remove alternate conformations. Moreover, it will remove water and ligand molecules, insert missing loop regions based on either SEQRES data or user specified loop definitions. Furthermore, it optimizes short and medium size loop regions with the LOOPER algorithm, minimizes the remaining loop regions, and calculates pK and protonates the structure.

Analysis of the crystal structure of SARS-CoV-2 spike receptor-binding domain bound to the ACE2 receptor revealed that there are three main areas the two proteins come in proximity. None is shown clearly as more important than the other. Preventing attachment of the two proteins will surely abort the entry of nCoV to the host cell by exploiting ACE2.

Herein, carnosine will be docked at the three points of attachment to reveal its orientation and binding patterns (“LibDock protocol” from Discovery Studio 2020, Biovia^®^). It will also be docked inside the active site of ACE2. Docking parameters were set as default except for the “docking preferences” changed to “high quality” and “minimize algorithm” changed to “Smart minimizer”. The latter performs 1000 steps of Steepest Descent with a RMS gradient tolerance of 3, followed by Conjugate Gradient minimization. The main amino acid in each of the three sites was used as a reference point for choosing the docking sphere and it was given to have a radius of 10 Å to give carnosine freedom to choose the best place and pose within the attachment site. The three amino acids that were assigned as reference for docking are Tyr41, His34, and Leu79.

Finally, searches on PubMed were conducted to check the current state of knowledge about ACEI in the COVID-19 infection. The following strategies for ACEI and COVID-19 were used: ((enalapril)[Title] OR (ramipril)[Title] OR (captopril)[Title] OR (trandolapril)[Title] OR (benazepril)[Title] OR (temocapril)[Title] OR (perindopril)[Title] OR (fosinopril)[Title] OR (ACE inhibitor)[Title]) AND ((COVID)[Title] OR (nCoV)[Title]).

Moreover, searches on PubMed were performed for carnosine and salicyl-carnosine using the following strategy: ((carnosine) OR (salicyl-carnosine)) AND ((COVID)[Title] OR (nCoV)[Title]).

Proof of concept for this study should follow with in vitro or animal model experimentation and that will be our subsequent future step.

### Statistical Analysis

Taking all clusters and elements in SwissDock, data for both full fitness and estimated ΔG is not normally distributed. Therefore, we used nonparametric tests to compare the three groups of carnosine, enalapril, and melatonin. Sample sizes were 253, 257, and 257 elements, respectively, and these were determined by the Swissdock engine. Of course, they represented the population of poses for each drug. We used the Kruskal–Wallis test as a nonparametric alternative to the one way ANOVA test and Mann–Whitney U test as an alternative to Student’s *t*-test. At the two-sided, alpha level of significance of 0.05, we generated *p*-values to compare the three groups.

## 4. Results and Discussion

As detailed in Section 3, the general Google search led us to the general scaffold structure of ACE2 inhibitors from Dales and Torres (Figure 2). The simplest form of this structure when R^1^ and R^3^ groups are replaced with hydrogen atoms, and R^2^ is dropped served as our reference in all subsequent bioactivity scoring and chemical similarity searches.

### 4.1. ACE Inhibitors and COVID-19

Clinicians are left in doubt about whether to continue or hold ACEI in COVID-19 patients. Currently, the consensus is there is no evidence it is harmful to continue ACEI [27]. Comparing the molinspiration bioactivity scores, noACEI is perfectly similar to the general structure of ACE2 inhibitors from Dales and Torres (Table 1).

However, on the basis of total absolute difference from the scaffold general structure of the various ligands, one would predict that enalapril could be the best ACEI to use for patients with COVID-19 (i.e., lowest absolute difference) followed possibly by ramipril (Table 2). ACEI similarity with the scaffold general structure was the highest for enalapril considering AP Tanimoto and MCS Min scores (Table 2). Although it would vary with the similarity scoring method, combining molinspiration bioactivity scores and similarity search, one would expect that enalapril is the best ACEI to use or continue in patients with COVID-19.

### 4.2. Carnosine Preliminary Molecular Docking

Carnosine had the highest similarity with the general structure of ACE2 inhibitor, shown to be about 82.7% (DrugBank, Alberta, Canada), 68.4% (MCS Tanimoto from ChemMine, Riverside, CA, USA), or 81.3% (MCS Max from ChemMine) (Figure 4). The only other highly similar approved or investigational compounds are histidine and 1-methylhistidine with DrugBank, quoted a similarity of 79.6% and 71.1%, respectively (Figure 4). All other similar histidine or nonhistidine structures showed less than 70% similarity. Both carnosine and histidine are nonpharmacological natural supplements available for human medical and experimental use [28]. At this time, there are no reports about carnosine levels in COVID-19 patients. However, it is expected that those levels would be depleted under COVID-19 oxidative stress [29]. Carnosine activates cellular stress response and possess antioxidant properties in many studies [30,31]. Salicyl-carnosine is quite similar to the scaffold general structure, although less than carnosine in terms of both bioactivity scores and chemical similarity, however, it is not available for human use (Figure 4, Table 1 and Table 2). Salicyl-carnosine presents as a better option to bypass the serum *carnosinases* and hence evade the need for high dose oral carnosine [32]. Carnosine and salicyl-carnosine resemble the scaffold general structure and have quite matching figures. However, carnosine seems to be a better candidate according to bioactivity scores and hence was selected in SwissDock Server for preliminary docking to host ACE2 cocrystallized with nCoV spike protein. Moreover, carnosine may be used intranasally which directly instills the drug at primary site of COVID-19 infection and hence experimental validation can be performed on both high oral gavage and intranasal administrations [33]. Furthermore, there are now multiple transgenic mouse models incorporating human ACE2 which can be used during the experimental validation phase for carnosine [34]. Back to SwissDock, a total of 42 clusters and 253 elements all showed favorable binding. Carnosine has full fitness to ACE2 (−3416.80 kcal/mol) more optimal than enalapril (−3281.90 kcal/mol) and melatonin (−3365.10 kcal/mol). Estimated ΔG for carnosine (−6.29 kcal/mol) was a bit lower than enalapril (−7.42 kcal/mol) but only slightly less than melatonin (−6.57 kcal/mol). Figure 5 shows the distribution of clusters (elements) for full fitness and estimated ΔG for the three drugs. Readers can easily infer that carnosine had an overall better full fitness and estimated ΔG much than both enalapril and melatonin. All comparisons were statistically significant (*p* value < 0.001). Carnosine best pose showed parts of the viral spike protein ligand binding with ACE2 retained at 1.75 Å, and as result, both protein–protein interactions are vulnerable to inhibitory actions by carnosine in this model.

Clearly, carnosine is anticipated to be an inhibitor of the protein–protein or at least a good starting point to design such potent ACE2 inhibitors. Based on a previous assessment of other drugs including most ACEI such as ramipril (docking score −6.12 kcal/mol), one can infer that carnosine is a better inhibitor of ACE2 than chloroquine (docking score −5.53 kcal/mol) and hydroxychloroquine, (docking score of −5.99 kcal/mol) both initially implicated as good drug qualifiers for COVID-19 [35]. Moreover, results of preliminary docking of carnosine to ACE2 were comparable to those of melatonin (−6.57 kcal/mol), yet another candidate drug for COVID-19. However, carnosine figures of preliminary docking are in contrast slightly less than other identified ACE2 inhibitors by Chikhale et al. such as with anoside X with an estimated ΔG (−7.07 kcal/mol) and ashwagandhanolide (−6.50 kcal/mol) [36]. Nevertheless, what makes carnosine probably stand out is that it is a widely commercialized supplement and hence can easily be studied or mobilized in the fight against COVID-19.

### 4.3. Detailed Molecular Docking and Modeling

Expectedly, there are two possible mechanisms in which carnosine acts as an inhibitor of COVID-19. First, via inhibitory binding to the active site especially chelating zinc atom. Second, by preventing the interaction between nCoV spike protein with ACE2. Based on these assumptions, our research group has conducted a docking study to evaluate the binding pattern of carnosine to the ACE2 active site and another to investigate the binding of carnosine to the protein–protein interaction sites between the two proteins. Figure 6 shows the two proteins interaction surfaces based on the crystal structure 2AJF from PDB.

Clearly, there are three major attachment points between the two proteins. Therefore, we docked carnosine molecule to the three areas on the surface of ACE2 to evaluate its binding pattern and its docking scores (Figure 7). The protein was prepared using “Prepare protein” protocol, and then the prepared protein used by assigning the three major sites to be docked by selecting the key amino acids in these areas. LibDock protocol from Discovery Studio 2020 was utilized to perform the docking and the high-quality option was used to get the best results from the docking process.

Of the three sites used for carnosine docking, site number 3 scored the highest LibDock score with 92.40, while site 1 was ranked to be the second with 90.88 and site 2 scored the lowest with 81.40. This indicated that carnosine had preference to stay at site number 3 where it has the best interactions with the ACE2 receptor surface. Figure 8 shows a 2D diagram for the type of interaction carnosine performs with the amino acids positioned at site 3. Clearly, the imidazole ring of carnosine performs two hydrogen bonds with Asp350 and Leu351 as well as salt bridge interactions with Asp350 and Glu37. These interactions are considered the main contributors to the LibDock score. Moreover, the carboxylic acid moiety of carnosine performs ion dipole bond with Phe356 and hydrogen bond with Gly354.

On the other hand, the docking of carnosine inside the active site was done to investigate the binding pattern and its score. The LibDock score was 85.49 and as Figure 9 shows the 2D pattern of carnosine binding with the active site. Obviously and as expected, the carboxylic acid group performs coordinate interaction with the zinc atom in which the lone pair of electrons of the carboxylic acid is shared to the empty orbital of zinc atom. This coordinate binding is the most important contributor to the LibDock score and is considered as an essential feature in inhibitors binding with metalloenzymes. Moreover, the imidazole ring in the structure performs pi-pi stacking interactions with His401 while the terminal primary amino of carnosine performs both hydrogen bound with Arg514 and pi-cation with Tyr 515. Flexible docking while the two proteins are attached yielded no useful information (data not shown).

In summary, carnosine seems to prefer sites 3 and 1 more so than ACE2 active site while at the same token performs essential interactions of ACE2 inhibitors. As a result, it is expected that it will be an important supplement to take to the next level of evaluation for COVID-19 and that is in vitro kit testing and animal modeling.

### 4.4. Literature Evaluation

PubMed search strategy yielded only three publications with commentaries on using carnosine and salicyl-carnosine, suggested for patients with COVID-19 [37,38,39]. In these papers, carnosine and salicyl-carnosine have been repurposed based on mere clinical and/or laboratory grounds. Jindal et al. suggested carnosine use based on its profile in mitigating complications such as those seen in patients with COVID-19 [37]. Jindal et al. have noted that this compound had antiviral, antioxidant, anti-inflammatory, as well as cardio-, neuro-, and musclo-protective effects. On the other hand, Lopachev et al. have proposed salicyl-carnosine as a better alternative to carnosine on the basis that it is much more stable in human blood [38]. Moreover, Lopachev et al. have shed the light on its antioxidant, anti-inflammatory, and antiplatelet effects, which are well suited for COVID-19 complications. Hipkiss et al. have alluded to the fact that carnosine could be administered nasally and hence escape the attention of serum carnosinase [39].

Our preliminary molecular docking of carnosine added more rationale to recommend further detailed docking and experimental validation and testing for this relatively safe natural supplement in patients with COVID-19. Moreover, carnosine has an antiviral activity including those against Dengue and Zika viruses [40]. To further highlight the potential of carnosine, it has been shown it is also active in ameliorating lung injury associated with the swine flu [41]. Finally, the detailed molecular docking and modeling showed carnosine to prefer two binding sites at the protein–protein interaction surface while at the same time performing essential interactions at the ACE2 active site.

### 4.5. Bioavailability and Physiological Role of Carnosine and Salicylcarnosine

The main physiological action of Carnosine is its promising therapeutic effects in ameliorating oxidative-based diseases [42]. This effect is mediated by carnosine inhibitory actions on advanced glycoxidation end products and lipoxidation end products. COVID-19 is one such disease and Soto et al. make a case that the Angiotensin (1–7) (A (1–7)) has a major protective effect in COVID-19 oxidative damage and lung injury [43]. Upregulated ACE2 may be protective too, but it is also a major means of nCoV entry into host cells and this process results in its downregulation [12,13,44,45]. Gul et al. discussed the growing evidence that shifting the equilibrium to the A (1–7)/ACE2 receptor access would be more favorable [46]. In this context, carnosine presents its ACE2 inhibitory effects to be mainly towards the entry into host cells and handling oxidative stress possibly without affecting the beneficial A (1–7) axis [47]. However, bioavailability of carnosine is a big concern due to the effect of inactivation by serum carnosinases. This could be overcome with new carnosine alternatives such as salicylcarnosine, nasal administration, or by supplementation of precursor molecules such as beta-alanine [38,39,48]. All the referenced studies above demonstrate beneficial effects for carnosine supplementation or augmentation in diseases that are well-known COVID-19 comorbid populations such as diabetic, hypertensive, kidney, brain, and muscular diseases.

### 4.6. Limitations

There are several important limitations for this chemical analysis. First, similarity in the chemical structure may sometimes fail to translate into clinical similarity. Yet, the fact that the general ACE2 inhibitors structure is quite close to carnosine could at least be the starting point to make new more matching lead molecules. Second, the scaffold general structure and the structures of carnosine, salicyl-carnosine, and histidine have several chiral centers. However, this can be simply taken into consideration while synthesizing or testing such molecules in further studies. Third, even the recommended scaffold and other structures by Dales and Torres may fail in the preclinical and clinical phases of research. Nevertheless, the urgency to find quick answers for the current COVID-19 pandemic is far from being a luxury anyway, and only real experimental validation in vitro, animal models, and patients can prove or disprove these drugs for the management of SARS-CoV-2. On the other hand, this research skipped in vitro or animal model validation due to the lockdowns in our country, Jordan, and the extra expense and expertise needed for these steps. Moreover, flexible docking yielded no useful information. However, in an unusual emergency such as the one the global community is experiencing with COVID-19, human use of carnosine, a widely available supplement, may well actually precede in vitro and animal validation. Finally, multiple receptor binding domain (RBD) variants with nCoV exist [49,50,51]. These may modify the nature, stability, and affinity of the protein–protein interaction. Current research had limited time to study all of them but the promising results we had should prompt extensive evaluation of carnosine or any future ACE2 inhibitors with such variants.

## 5. Conclusions

This is the first, to our best knowledge, preliminary and detailed molecular docking and modeling study that shows carnosine has probably excellent fit with an inhibitor activity against ACE2 and its protein–protein interaction with the viral spike protein. If carnosine or its derivatives live through the test of time to be shown effective on COVID-19, this study would prove that methods provided could be used by practitioners for proposing and solving other disastrous health patient-centered challenges.

## Figures and Tables

**Figure 1 molecules-25-05605-f001:**
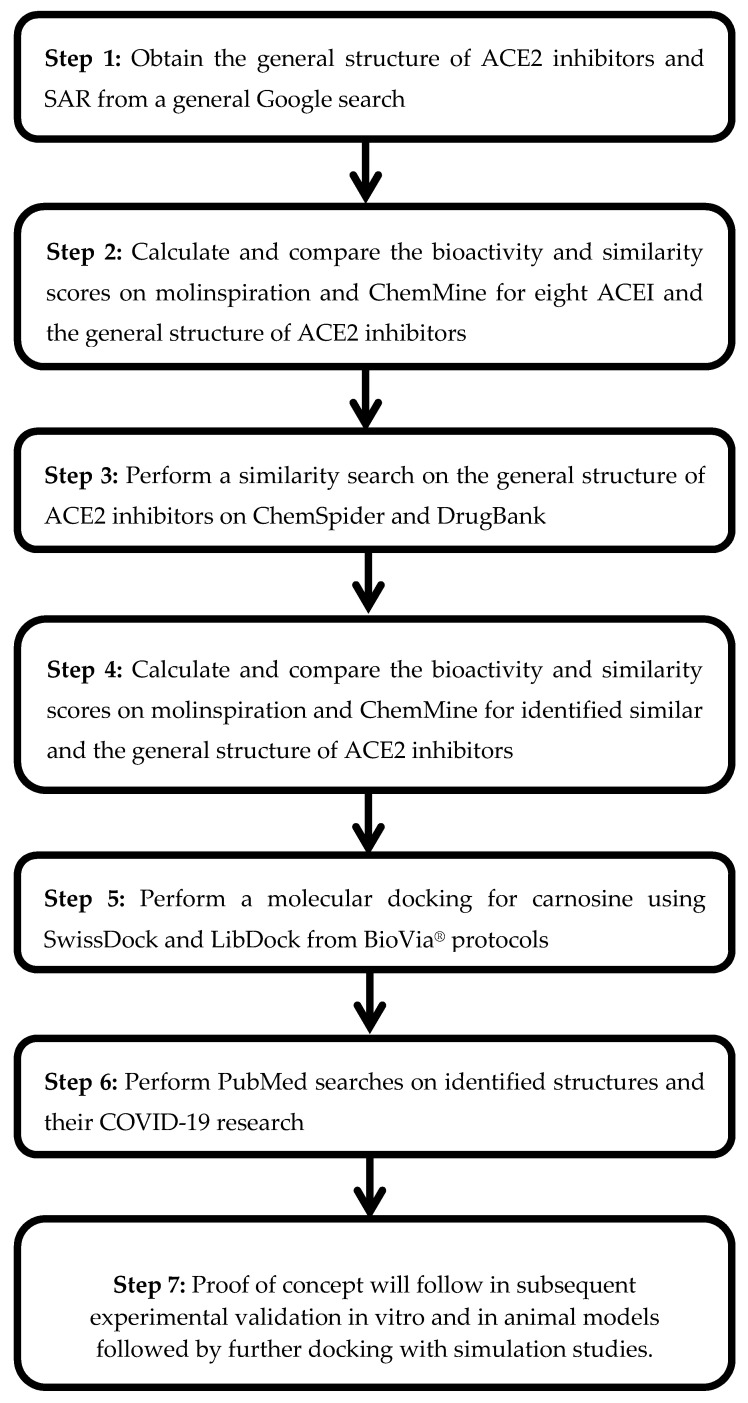
Overview of the study methodology.

**Figure 2 molecules-25-05605-f002:**
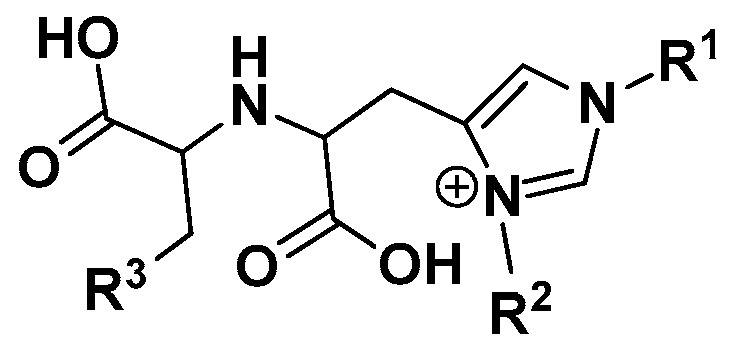
Scaffold general structure for Angiotensin Converting Enzyme 2 ACE2 inhibitors from Dales and Torres.

**Figure 3 molecules-25-05605-f003:**
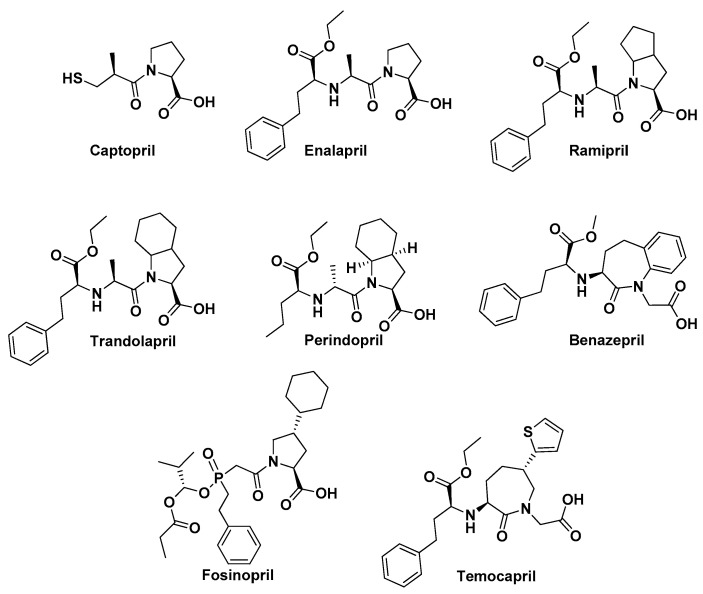
Structures for the eight selected angiotensin-converting enzyme inhibitors.

**Figure 4 molecules-25-05605-f004:**
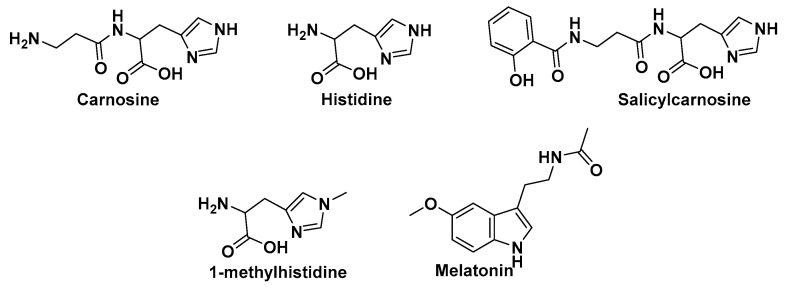
Chemical structures of new suggested molecules for COVID-19 treatment.

**Figure 5 molecules-25-05605-f005:**
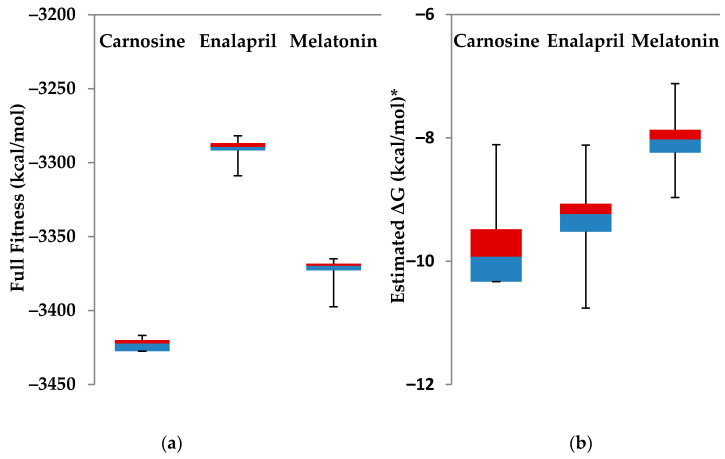
A full fitness (**a**) and an estimated ΔG (**b**) of all docking clusters to host ACE2 cocrystallized with nCoV viral spike protein. (* All comparisons are statistically significant at two-sided alpha level of significance of 0.05, all *p*-values < 0.001).

**Figure 6 molecules-25-05605-f006:**
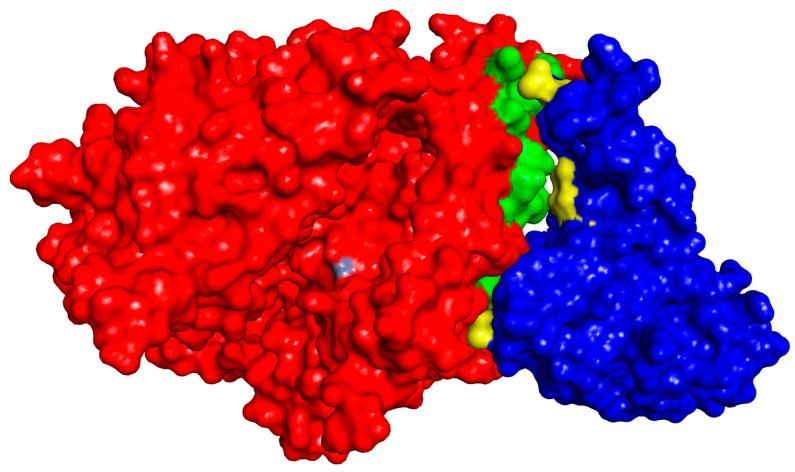
nCoV spike protein (blue) interacting with ACE2 (red). The yellow-green areas are showing the major attachment points between the two proteins. The grey area in ACE2 represents the zinc atom in the center of the active site.

**Figure 7 molecules-25-05605-f007:**
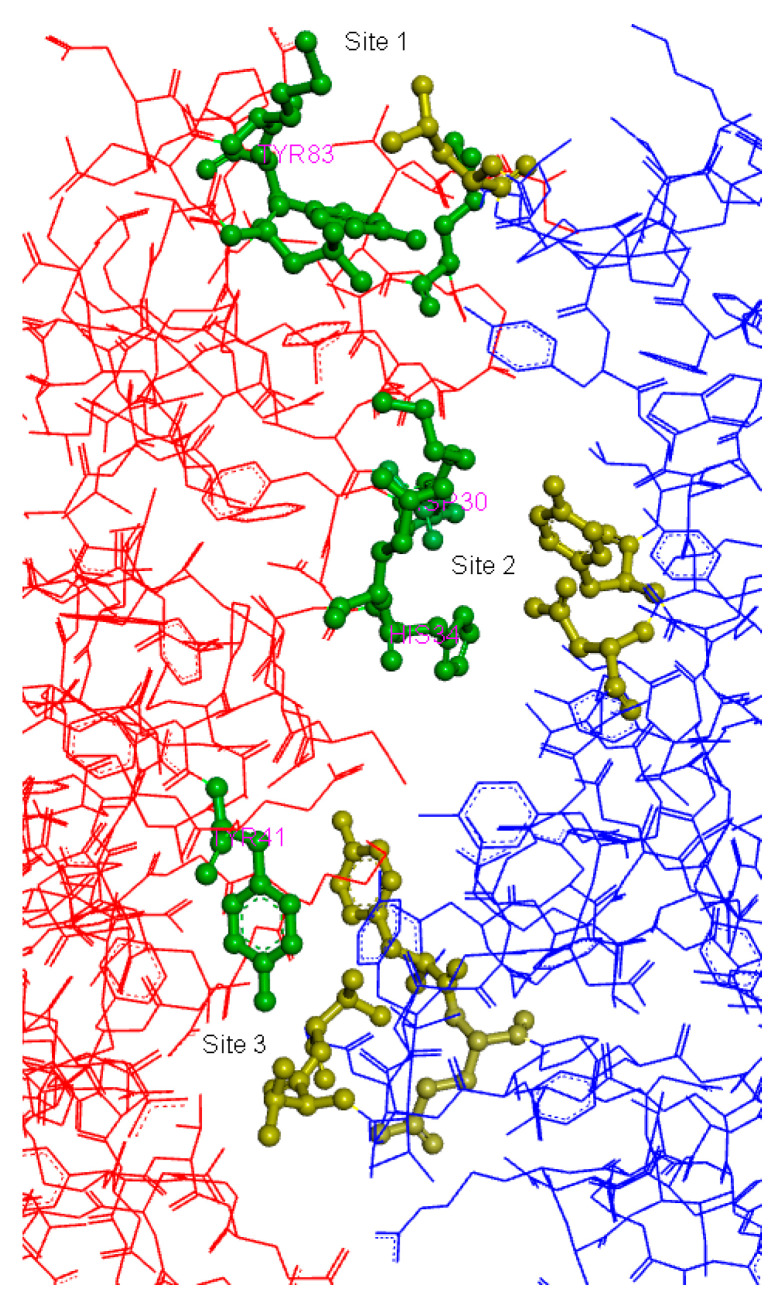
The three major interacting sites between the ACE2 (red) and nCoV (blue) showing each site docked with the major amino acids involved in the interaction.

**Figure 8 molecules-25-05605-f008:**
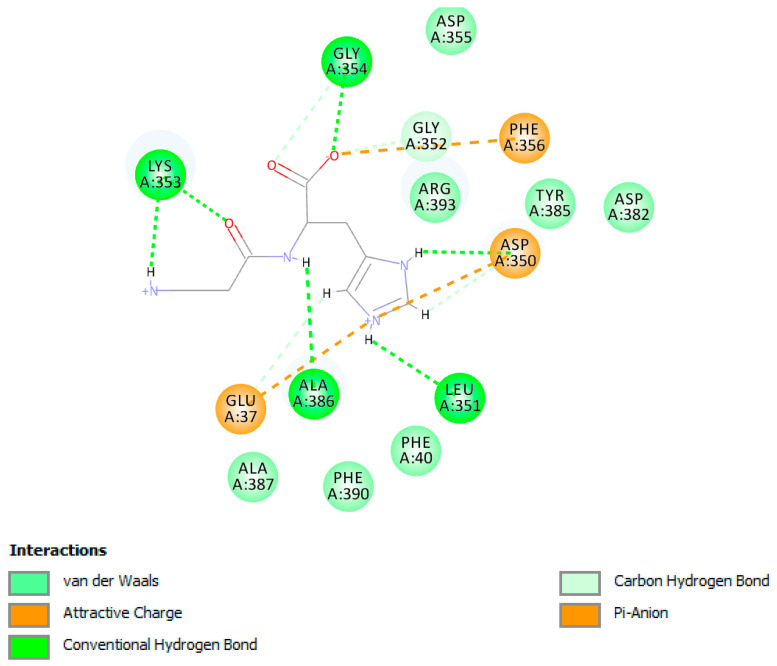
2D diagram shows the docked pose of the highest score carnosine within the surface of ACE2 at site 3.

**Figure 9 molecules-25-05605-f009:**
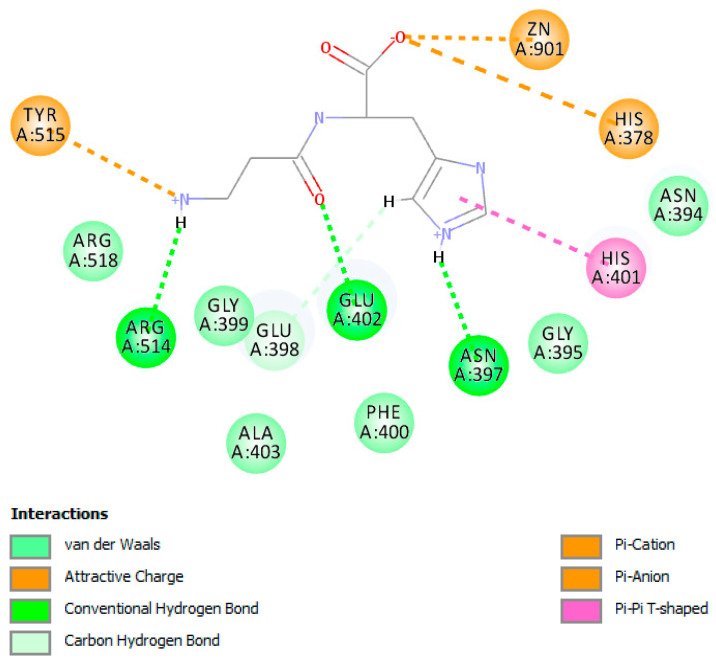
2D diagram shows the docked pose of the highest score carnosine within the active site of ACE2.

**Table 1 molecules-25-05605-t001:** Molinspiration bioactivity scores for the scaffold (general structure) of ACE2 inhibitors from Dales and Torres, ACEI, carnosine, and salicyl-carnosine.

Drug	1 *	2	3	4	5	6	Total Difference **
General scaffold	0.46	0.47	−0.15	−1.25	0.52	0.58	0
Captopril	−0.14	−0.08	−0.98	−0.55	0.97	0.50	3.21
Enalapril	0.36	0.16	−0.30	−0.08	0.70	0.18	2.31
Ramipril	0.36	0.08	−0.36	−0.12	0.78	0.23	2.44
Trandolapril	0.36	0.05	−0.44	−0.16	0.76	0.17	2.55
Perindopril	0.36	0.02	−0.52	−0.23	0.83	0.20	2.63
Benazepril	0.22	0.09	−0.35	0.07	0.43	0.10	2.71
Fosinopril	0.44	0.07	−0.31	−0.11	1.03	0.41	2.40
Temocapril	0.10	−0.13	−0.45	−0.21	0.49	0.03	2.88
Carnosine	0.61	0.48	−0.06	−1.2	0.65	0.73	0.58
Salicyl-carnosine	0.61	0.26	0.08	−0.58	0.63	0.49	1.46

* 1: GPCR ligand 2: Ion Channel Modulator 3: Kinase Inhibitor 4: Nuclear Receptor Ligand 5: Protease Inhibitor 6: Enzyme Inhibitor. ** Total sum of absolute difference at each of the six receptors between the drug and the general scaffold structure value.

**Table 2 molecules-25-05605-t002:** ChemMine similarity scores for the scaffold (general structure) of ACE2 inhibitors, ACEI, carnosine, and salicyl-carnosine.

Drug	AP Tanimoto	MCS Tanimoto	MCS Size	MCS Min	MCS Max
Scaffold	-	-	-	-	-
Captopril	0.128	0.429	9	0.643	0.563
Enalapril	0.169	0.344	11	0.688	0.407
Ramipril	0.147	0.314	11	0.688	0.367
Trandolapril	0.138	0.306	11	0.688	0.355
Perindopril	0.138	0.355	11	0.688	0.423
Benzapril	0.114	0.306	11	0.688	0.355
Fosinopril	0.051	0.196	9	0,.563	0.231
Temocapril	0.110	0.297	11	0.688	0.344
Carnosine	0.364	0.684	13	0.813	0.813
Salicyl-carnosine	0.214	0.464	13	0.813	0.520

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
