# Peer review of "Carnosine to Combat Novel Coronavirus (nCoV): Molecular Docking and Modeling to Cocrystallized Host Angiotensin-Converting Enzyme 2 (ACE2) and Viral Spike Protein"

_molecules, 2020, doi:10.3390/molecules25235605_

Round 1

Reviewer 1 Report

The subject of the article is interesting and presented research is properly designed. However, the urgent need for developing COVID-19 ameliorating compounds, had resulted with a great number of similar work. Nonetheless, carnosine due to its high availability and safety in supplementation seems to be a molecule of high potential.

Having said that, the manuscript ought to be improved in following aspects:

Major Issues:

1) Major English language and style edditing required 

  •  especially in lines (146-154), (168-173), (179-185), (206-209), (255), (269-277) and (318-322) the grammar, style and clarity ought to be improved and appropriate scientific language used
  •  Usage of adequate phrases and punctuation in the whole manuscript body should be corrected. Preferably by English native speaker

2) The authors presented molecular docking results for carnosine to the three areas on the surface of ACE2.

  • The manuscript would greatly benefit if the molecular docking simulations would be performed for carnosine with and without the presence of an adequate spike protein fragment at each of the three ACE2 binding sides and the binding energy of these variants should be compared.  
  • Detailed analysis of intermolecular contacts (ICs) and list of amino acids which are present in the protein–protein interaction sites of spike protein fragment with ACE2 receptor in absence and presence of carnosine should be carried out and presented

3) Authors should put more attention to bioavailability of carnosine or salicyl-carnosine. Describe the role of this molecules in physiology and perhaps discuss the usage of carnosine precursors such as beta-alanine which, long term supplementation with relation to carnosine levels are well documented.

Minor issues:

1) Lines 184-185, please specify "the main amino acids in each of the three sites" that were used as a reference point

2) Figure 5. The level of statistical significance should be given for each molecule, and performed statistical analysis described

3) lines 284 - 286, providing the ACEI values for other described molecules

Reviewer 2 Report

In this manuscript, the authors conducted chemo-bioinformatics and molecular docking studies of angiotensin-converting enzyme 2 (ACE2), which is a host receptor against SARS-CoV-2. The chemo-bioinformatics analysis revealed that carnosine is a promising ACE2 inhibitor candidate. Furthermore, the in silico molecular docking between ACE2 and carnosine together with ACEI inhibitors proposed their interaction mode at atomic level. The presented results would provide valuable insights for developing new anti-COVID-19 drugs. This reviewer would recommend the manuscript for publication in Molecules, if the following minor point is adequately addressed.

Minor point:

To date, structures of SARS-CoV-2 S protein complexed with ACE2 have been determined not only by X-ray crystallography but also by cryo-electron microscopy demonstrating their interaction modes at atomic level. Nevertheless, the ACE2/SARS-CoV-2 S protein complex is denoted only as “co-crystallized structure” throughout the manuscript including the title. This reviewer wonders why the authors are sticking to such a restricted phrase, even though the present docking was performed based on a co-crystallized structure.  

Reviewer 3 Report

The authors have presented a computational study including molecular docking of several drugs to identify carnosine as a promising candidate for providing inhibitory interactions to destabilize hACE2-spike complex (which is deemed as step 0 in covid19 pathogenesis).

This study, even though provides prior for experimental validation, there are some real concerns regarding several aspects of the study.

In terms of technicality, the authors refer to the virus (and its spike) as covid19 (see Fig 7). Covid19 actually, happens to be the name of the disease and SARS-CoV-2 the strain of the virus. The authors seem to use this interchangeably - which needs to be fixed across the manuscript.

It is important to note that there has been >15 manuscripts till date that have explained the biophysical underpinnings SARS-CoV-2 and ACE2 interactions. So this is already well understood and not a novel reporting as the authors claim in line 55 : "ACE2 seems to be a good, less evaluated, starting point for potential COVID-19 therapies."

The authors need to extend their docking studies on the various hACE2 and RBD variants that improve human infectivity reported in the following manuscripts:

1) Fig 6: https://www.biorxiv.org/content/10.1101/2020.06.17.157982v1

2) Table 2: https://www.sciencedirect.com/science/article/pii/S2001037020304037

3) https://science.sciencemag.org/content/369/6508/1261

This would be interesting to know whether carnosine can still destabilize the spike-hACE2 complex for the RBD variants, and human ACE2 genotypic variants that makes humans more susceptible to covid19.

The second major problem I have with the manuscript is: in line 354 the authors mention that there remains room for more "detailed docking" to infer more biophysical insights. I totally agree with the authors and I think they must include those in the revised version of the manuscript to make a case for this article to be published in Molecules.

ZINC, Z-DOCK 3, RosettaDock are known to handle the role of implicit solvation at protein interfaces. It is strongly advised that the authors use the non-server options for these dockers so as to be able to comment more about the choice of molecular mechanical parameters for the docking. These would be crucial to reproduce the results. Supplementary information should be provided about the partial charges on the atoms, spring constants for the bonds of the drug molecule, improper dihedrals, and importantly the information about which bond-angle torsions were permitted during docking and which were not.

Finally, I was excited to see validation on animal models and in vitro testing in Flowchart 1. However, I was disappointed to not find any original results on that front, and instead, there were literature-based evidences to support carnosine dosage. Now I am confused about what extra experimental validation can the author's add in case they are planning to follow this up with another manuscript. Also, that puts the novelty of this article to question.

This now, reads like a manuscript where the authors have re-validated an already experimentally promising drug and done docking to report that it works (which is intuitive). 

The authors should also comment on the fact that they are advising ACE-inhibitors while there has been several clinical reports (in Lancet, Science, and Nature) that have said otherwise. ARBs have shown more promise as they stabilize the human ATR1-ACE2 transmembrane complex thus making ACE2 unavailable for spike binding, and unlike ACEi - they DO NOT increase covid19 susceptibility to individuals with high blood pressure.

The authors should revise this manuscript substantially for subsequent consideration.

Round 2

Reviewer 1 Report

All reised major and minor points were addressed and answered. The Manuscript language was corrected and sufficient explanation was provided for major point 2 and 3. Overall, readability of the manuscript was improved. Modifications to Figure 1 and adding the section 4.4. were also helpful.   

Reviewer 3 Report

Given the financial and other logistic constraints (that the author cites) during these special times, I guess there is no room to ask for more validation on this manuscript.

I think this paper can be accepted as is, even though it raises some doubts about the priority claims made in the paper. But we can only hope that there will be a follow-up soon which will complete the story.